# Entrywise Convergence of Iterative Methods for Eigenproblems

**Vasileios Charisopoulos**
Department of Operations Research & Information Engineering
Cornell University
Ithaca, NY 14853
vc333@cornell.edu

**Austin R. Benson**
Department of Computer Science
Cornell University
Ithaca, NY 14853
arb@cs.cornell.edu

**Anil Damle**
Department of Computer Science
Cornell University
Ithaca, NY 14853
damle@cornell.edu

## Abstract

Several problems in machine learning, statistics, and other fields rely on computing eigenvectors. For large scale problems, the computation of these eigenvectors is typically performed via iterative schemes such as subspace iteration or Krylov methods. While there is classical and comprehensive analysis for subspace convergence guarantees with respect to the spectral norm, in many modern applications other notions of subspace distance are more appropriate. Recent theoretical work has focused on perturbations of subspaces measured in the $\ell_{2\to\infty}$ norm, but does not consider the actual computation of eigenvectors. Here we address the convergence of subspace iteration when distances are measured in the $\ell_{2\to\infty}$ norm and provide deterministic bounds. We complement our analysis with a practical stopping criterion and demonstrate its applicability via numerical experiments. Our results show that one can get comparable performance on downstream tasks while requiring fewer iterations, thereby saving substantial computational time.

## 1 Introduction & Background

Spectral methods play a fundamental role in machine learning, statistics, and data mining. Methods for foundational tasks such as clustering [52]; semi-supervised learning [36]; dimensionality reduction [5, 22, 45]; latent factor models [25] ranking and preference learning [38, 51]; graph signal processing [42, 49]; and covariance estimation all use information about eigenvalues and eigenvectors (or singular values and singular vectors) from an underlying data matrix (either directly or indirectly). The pervasiveness of spectral methods in machine learning applications[1] has greatly influenced the last decade of research in large-scale computation, including but not limited to sketching / randomized NLA [26, 37, 53] as well as theoretical guarantees for linear algebra primitives (*e.g.*, eigensolvers, low-rank decompositions) in previously overlooked settings.

In many of these cases, the relevant information is in the "leading" eigenvectors, *i.e.,* those corresponding to the $k$ algebraically largest eigenvalues for some $k$ (possibly after shifting and rescaling). To avoid performing a full eigendecomposition, these are typically approximated with iterative algorithms such as the power or Lanczos methods. The approximation quality, as measured by

subspace distance (equivalent to using the $\ell_2$ norm, up to rotation), is well-understood and enjoys comprehensive convergence analysis [16, 24, 43, 47].

While spectral norm error analysis has been the standard-bearer for numerical analysis, recent work has considered different subspace distance measures [8, 11, 19, 54]. The motivation for these changes is statistical, as opposed to numerical: we observe a matrix $\tilde{A} = A + E$, where $E$ is a source of noise and $A = \mathbb{E}\left[\tilde{A}\right]$ is the "population" version of $A$, containing the desired spectral information. We are then interested in $\|\tilde{u}_i \pm u_i\|_\infty$ as a distance measure between the eigenvectors of $\tilde{A}$ and $A$. Here, the $\ell_\infty$ norm captures "entry-wise" error and is more appropriate when we care about maximum deviation; for example, when entries of the eigenvector are used to rank nodes or provide cluster assignments in a graph. This type of distance is often much smaller than the spectral norm and, in contrast to the latter, reveals information about the distribution of error over the entries. Recent theoretical results relate the noise $E$ to the perturbation in the eigenvectors, as measured by $\ell_\infty$ or $\ell_{2\to\infty}$ norm errors [10, 15, 20, 28, 31]. Moreover, these results are often directly connected to machine learning problems [1, 12, 17, 56].

The message from this body of literature is that when eigenvectors are interpreted entry-wise, we should measure our error entry-wise as well. The aforementioned works show what we can do *if* we have eigenvectors satisfying perturbation bounds in a different norm, but do not address their *computation*. Numerical algorithms typically use the $\ell_2$ norm, yet the motivation for norms like $\ell_{2\to\infty}$ is that $\ell_2$ can be a severe overestimate for the relevant approximation quality. Moreover, despite the long history of research into stopping criteria for iterative methods in the unitarily-invariant setting [3, 4, 6, 23, 29], there are no generic stopping criteria closely tracking the quality of an approximation in the $\ell_{2\to\infty}$ norm. For example, downstream tasks that depend on entrywise ordering, such as graph bipartitioning via the (approximate) Fiedler vector [18] or spectral ranking via the *Katz* centrality [40] employ $\ell_2$ bounds, when instead the $\ell_\infty$ norm would constitute a better proxy. Some local spectral graph partitioning methods can be written as iteratively approximating an eigenvector in a (scaled) $\ell_\infty$ norm [2], but these algorithms are far more specialized than general eigensolvers. The situation is similar when using more than one eigenvector; in spectral clustering with $r$ clusters, after an appropriate rotation of the eigenvector matrix, the magnitude of the elements in the $i^{\text{th}}$ row measures the per-cluster membership likelihood of the $i^{\text{th}}$ node, making the $\ell_{2\to\infty}$ norm (which is invariant to unitary transformations on the right) a more appropriate distance measure than the spectral norm (see *e.g.*, [34]).

Here, we bridge this gap by providing an analysis for the convergence of subspace iteration, a widely-used iterative method for computing leading eigenvectors, in terms of $\ell_{2\to\infty}$ errors. We complement that with a practical stopping criterion applicable to any iterative method for invariant subspace computation that tracks the $\ell_{2\to\infty}$ error of the approximation. Our results show how, for a given error tolerance, one can perform many fewer subspace iterations to get the same desired performance on a downstream task that uses the eigenvectors (or, more generally, an invariant subspace) — as $\|V\|_{2\to\infty} \in [1, \sqrt{r}] \max_{i,j} |V_{ij}|$ for $V \in \mathbb{R}^{n\times r}$, and often $r \ll n$, our bounds are also a good "proxy" for the maximum entrywise error. The aforementioned reduction in iterations directly translates to substantial reductions in computation time. We demonstrate our methods with the help of applications involving real-world graph data, including node ranking in graphs, sweep cut profiles for spectral bipartitioning, and general spectral clustering.

## 1.1   Notation

We use the standard inner product on Euclidean spaces, defined by $\langle X, Y \rangle := \text{Tr}\left(X^\mathsf{T} Y\right)$ for vectors/matrices $X, Y$. We write $\mathbb{O}_{n,k}$ for the set of matrices $U \in \mathbb{R}^{n\times k}$ such that $U^\mathsf{T} U = I_k$, dropping the second subscript when $n = k$. We use standard notation for norms, namely $\|A\|_2 := \sup_{x:\|x\|_2=1} \|Ax\|_2$ and $\|A\|_F := \sqrt{\langle A, A \rangle}$. Moreover, we remind the reader that the $\ell_\infty \to \ell_\infty$ operator norm for a matrix $A \in \mathbb{R}^{m\times n}$ is given by $\|A\|_\infty := \max_{i\in[m]} \|A_{i,:}\|_1$, where $A_{i,:}$ denotes the $i^{\text{th}}$ row of $A$ and $A_{:,i}$ denotes its $i^{\text{th}}$ column. Finally, the $\ell_{2\to\infty}$ norm is defined by

$$\|A\|_{2\to\infty} := \sup_{x:\|x\|_2=1} \|Ax\|_\infty = \max_{i\in[m]} \|A_{i,:}\|_2. \tag{1}$$

**Subspace distances.** Given two orthogonal matrices $V, \tilde{V} \in \mathbb{O}_{n,r}$ inducing subspaces $\mathcal{V}, \tilde{\mathcal{V}}$, their so-called subspace distance is defined as $\text{dist}_2(V, \tilde{V}) := \|VV^\mathsf{T} - \tilde{V}\tilde{V}^\mathsf{T}\|_2$, with several equivalent

---

**Algorithm 1** Subspace iteration

---

**Input**: initial guess $Q_0 \in \mathbb{O}_{n,k}$, symmetric matrix $A$, iterations $T$
**for** $t = 1, 2, \ldots, T$ **do**
　　$V^{(t)} := AQ_{t-1}; \quad Q_t, R_t = \mathtt{qr}(V^{(t)})$　　　　　　　　　　▷ QR decomposition
**end for**
**return** $Q_T$

---

definitions, *e.g.*, via the concept of *principal angles*, or via $\left\| V_\perp^\mathsf{T} \tilde{V} \right\|_2$, where $V_\perp$ is a basis for the subspace orthogonal to $\mathcal{V}$. Here we will use a slightly different notion of distance between subspaces with respect to $\|\cdot\|_{2\to\infty}$ defined as

$$\mathrm{dist}_{2\to\infty}(V, \tilde{V}) := \inf_{O \in \mathbb{O}_{r,r}} \left\| V - \tilde{V}O \right\|_{2\to\infty}. \tag{2}$$

This metric allows us to control errors in a "row-wise" or "entry-wise" sense; for example, in the case where $r = 1$ this reduces to infinity norm control over the differences between eigenvectors. Finally, some of the stated results use the *separation between matrices* measured along a linear subspace (with respect to some norm $\|\cdot\|_\star$):

$$\mathsf{sep}_{\star, W}(B, C) = \inf \left\{ \|ZB - CZ\|_\star \mid \|Z\|_\star = 1, Z \in \mathrm{range}(W) \right\} \tag{3}$$

When $\|\cdot\|_\star$ is unitarily invariant and $B, C$ are diagonal, we recover $\mathsf{sep}_{\star, W}(B, WCW^\mathsf{T}) = \lambda_{\min}(B) - \lambda_{\max}(C)$; thus $\mathsf{sep}$ generalizes the notion of an eigengap.

## 2　Convergence of subspace iteration

In this section, we analyze the convergence of subspace iteration (Algorithm 1) with respect to the $\ell_{2\to\infty}$ distance. In particular, we assume that we are working with a symmetric matrix $A$ with eigenvalue decomposition

$$A = V\Lambda V^\mathsf{T} + V_\perp \Lambda_\perp V_\perp^\mathsf{T}, \tag{4}$$

where $\Lambda, \Lambda_\perp$ are diagonal matrices containing the $r$ largest and $n - r$ smallest eigenvalues of $A$. For simplicity, we assume that the eigenvalues satisfy $\lambda_1(A) \geq \cdots \geq \lambda_r(A) > \lambda_{r+1}(A) \geq \ldots \lambda_n(A)$ and, furthermore, that $\min_{k=1,\ldots,r} |\lambda_k(A)| > \max_{k=r+1,\ldots,n} |\lambda_k(A)|$.[2]

Our perturbation bounds and stopping criterion both involve the *coherence* of the principal eigenvector matrix, which is a standard assumption in compressed sensing [9].

**Definition 1** (Coherence). *Given $V \in \mathbb{O}_{n,r}$, we define its **coherence** as the smallest $\mu > 0$ such that*

$$\|V\|_{2\to\infty} = \max_{i \in [n]} \|VV^\mathsf{T} e_i\|_2 \leq \mu \sqrt{\frac{r}{n}}. \tag{5}$$

Given Definition 1, a matrix of eigenvectors is *incoherent* if none of its rows have a large element (i.e. all elements are on the order of $\sqrt{1/n}$).

The following result shows that $\mathrm{dist}_{2\to\infty}(Q_t, V)$ can be considerably smaller than $\mathrm{dist}_2(Q_t, V)$. Unfortunately, our analysis involves the unwieldy term $\|V_\perp \Lambda_\perp^t V_\perp^\mathsf{T}\|_\infty$, which is nontrivial to upper bound to obtain a better rate than that obtained using norm equivalence. To circumvent this, we impose a technical assumption.

**Assumption 1.** *For the matrix of interest, $V_\perp$ satisfies*

$$\|V_\perp \Lambda_\perp^t V_\perp^\mathsf{T}\|_\infty \leq C \cdot \lambda_{r+1}^t \|V_\perp V_\perp^\mathsf{T}\|_\infty, \tag{6}$$

*for a small constant $C$ and all powers $t \in \mathbb{N}$.*

Assumption 1 arises due to our proof technique, and may be removed by a more careful analysis (the supplement contains a preliminary result in this direction). We *empirically verified that it holds* with a constant $C < 2$, for all powers $t$ up to the last elapsed iteration of Algorithm 1 in our numerical experiments of Section 4; this makes us rather confident that it is a reasonable assumption in real-world datasets.

**Proposition 1.** *Suppose Assumption 1 holds. The iterates $\{Q_t\}$ produced by Algorithm 1 with initial guess $Q_0$ satisfy*

$$\mathrm{dist}_{2\to\infty}(Q_t, V) \leq \left(\frac{\lambda_{r+1}}{\lambda_r}\right)^t \left[\mu\sqrt{\frac{2r}{n}}\frac{d_0}{\sqrt{1-d_0^2}} + \frac{C(1+\mu\sqrt{r})}{\sqrt{1-d_0^2}}\mathrm{dist}_{2\to\infty}(Q_0, V)\right], \qquad (7)$$

*where $d_0 := \|Q_0^\mathsf{T} V_\perp\|_2 \equiv \mathrm{dist}_2(Q_0, V)$, $r = \dim(V)$, and $\mu$ is the coherence of $V$.*

*Proof.* The proof is attached in the supplementary material. □

When $\lambda_{r+2} \ll \lambda_{r+1}$, a slight modification of the above proof yields a potentially refined upper bound. The proof is contained in the supplementary material as well.

**Proposition 2.** *The iterates $\{Q_t\}$ produced by Algorithm 1 with initial guess $Q_0$ satisfy*

$$\begin{aligned}
\mathrm{dist}_{2\to\infty}(Q_t, V) \leq{} & \left(\frac{\lambda_{r+1}}{\lambda_r}\right)^t \left[\mu\sqrt{\frac{2r}{n}}\cdot\frac{d_0}{\sqrt{1-d_0^2}} + \frac{\|v_{r+1}v_{r+1}^\mathsf{T}\|_\infty}{\sqrt{1-d_0^2}}\cdot\mathrm{dist}_{2\to\infty}(Q_0, V)\right] \\
& + \left(\frac{\lambda_{r+2}}{\lambda_r}\right)^t \frac{d_0}{\sqrt{1-d_0^2}},
\end{aligned} \qquad (8)$$

*where $\mu$ is the coherence of $V$.*

Typically, we expect that $\mathrm{dist}_{2\to\infty}(Q_0, V) \ll \mathrm{dist}_2(Q_0, V)$, since otherwise the error is highly localized in just a few rows of the matrix. Therefore, Propositions 1 and 2 show that we can achieve significant practical improvements in that regime (recall that convergence analysis with respect to the spectral norm gives a rate of $\left(\lambda_{r+1}/\lambda_r\right)^t \frac{d_0}{\sqrt{1-d_0^2}}$ [24]). Section 4 illustrates this concept in practical examples.

## 3 Stopping criteria

In this section, we propose and analyze a stopping criterion for tracking convergence with respect to the $2 \to \infty$ norm. Notably, this stopping criterion is generic and applicable to any iterative method for computing an invariant subspace.[3] Suppose that we have

$$AQ - QS = E, \quad \|E\|_2 \leq \varepsilon, \quad Q \in \mathbb{O}_{n,r}, \quad S = S^\mathsf{T}.$$

Then it is well-known [24, Theorem 8.1.13] that there exist $\mu_1, \ldots, \mu_r \in \Lambda(A)$ such that $|\mu_k - \lambda_k(S)| \leq \sqrt{2}\varepsilon, \quad \forall k \in [r]$. This provides a handy criterion for testing convergence of eigenvalues, by setting $S = D_t$, the diagonal matrix of approximate eigenvalues at the $t^{\text{th}}$ step and $Q = Q_t$, the orthogonal matrix of approximate eigenvectors. The following lemma is straightforward to show.

**Lemma 1.** *Suppose that $A = A^\mathsf{T} \in \mathbb{R}^{n\times n}$ satisfies $AQ - QS = E$, $Q \in \mathbb{O}_{n,r}$, for some **diagonal** matrix $S$. Then $Q$ is an invariant subspace of the matrix $A - EQ^\mathsf{T}$.*

We demonstrate that checking $\|AQ - QS\|$ leads to an appropriate stopping criterion for iterative methods, and simplifies under standard incoherence assumptions. The proof of Proposition 3, crucially relies on a perturbation bound from [15] and is deferred to the supplement.[4]

**Proposition 3.** *Assume that $A$ is symmetric with $V$ as its dominant subspace and $V_\perp$ spans the orthogonal complement of $V$, with $V \in \mathbb{O}_{n,r}$; furthermore, suppose that $A$ satisfies the conditions of Lemma 1 for some $Q$ and let $\mathtt{gap} := \min\left\{\lambda_r(A) - \lambda_{r+1}(A), \mathtt{sep}_{(2,\infty),V_\perp}(\Lambda, V_\perp\Lambda_\perp V_\perp^\mathsf{T})\right\}$. Then, if $Q$ is the leading invariant subspace of $A - EQ^\mathsf{T}$ and $\|E\|_2 \leq \frac{\mathtt{gap}}{5}$, we have*

$$\mathrm{dist}_{2\to\infty}(V, Q) \leq 8\|V\|_{2\to\infty}\left(\frac{\|E\|_2}{\lambda_r - \lambda_{r+1}}\right)^2 + 2\|V_\perp V_\perp^\mathsf{T}\|_\infty \frac{\|E\|_{2\to\infty}}{\mathtt{gap}}\cdot\left(1 + \frac{2\|E\|_2}{\lambda_r - \lambda_{r+1}}\right).$$

**Corollary 1.** *Suppose that $V \in \mathbb{O}_{n,r}$ with coherence $\mu$ and that the conditions of Lemma 1 are satisfied with $\|E\|_2 \leq \varepsilon_1$, $\|E\|_{2\to\infty} \leq \varepsilon_2$. Then the approximate eigenvector matrix $Q$ satisfies*

$$\mathrm{dist}_{2\to\infty}(V, Q) \leq 8\mu\sqrt{\frac{r}{n}}\left(\frac{\varepsilon_1}{\lambda_r - \lambda_{r+1}}\right)^2 + 2\frac{1 + \mu\sqrt{r}}{\mathsf{gap}} \cdot \left(\varepsilon_2 + 2\frac{\varepsilon_1\varepsilon_2}{\lambda_r - \lambda_{r+1}}\right), \qquad (9)$$

*with* $\mathsf{gap}$ *defined as in Proposition 3.*

**Practical issues.** Checking the criterion of Corollary 1 requires computing $\|E\|_2$, $\|E\|_{2\to\infty}$ and estimating $\mathsf{gap}$. The first two terms are straightforward. To estimate $\mathsf{gap}$ in practice, we assume that $\mathsf{sep}_{2\to\infty,V_\perp}(\Lambda, V_\perp\Lambda_\perp V_\perp^\mathsf{T})$ is a small multiple of the $\lambda_r - \lambda_{r+1}$, motivated by the observation that $\mathsf{sep}_{2\to\infty}$ is *at worst* a factor of $\frac{1}{\sqrt{n}}$ smaller than the eigengap [15, Lemma 2.4]; moreover, this $\frac{1}{\sqrt{n}}$ factor is typically loose. To estimate $\lambda_r - \lambda_{r+1}$, we may use a combination of techniques, such as augmenting the "seed" subspace by a constant number of columns and setting $|\lambda_r - \lambda_{r+1}| \approx \hat{\lambda}_r - \hat{\lambda}_{r+1}$ – where $\hat{\lambda}_i = \lambda_i(Q^\mathsf{T}AQ)$ are the approximate eigenvalues – as it is well known that eigenvalue estimates converge at a quadratic rate for symmetric matrices [50].

In the absence of incoherence information, it is not possible to evaluate Equation (9), and we may instead replace all quantities in the residual by estimates (which is common practice for unknown quantities in standard eigensolvers). For any $B$, $\|BQ_t\|_{2\to\infty} \approx \|BV\|_{2\to\infty}$ (by [10, Prop. 6.6] and since $Q_tQ_t^\mathsf{T} \approx VV^\mathsf{T}$ after sufficient progress). Similar arguments for the other terms yield an approximated residual:

$$\mathsf{res}_{2\to\infty}(t) := 8\|Q_t\|_{2\to\infty}\left(\frac{\|E\|_2}{\lambda_r - \lambda_{r+1}}\right)^2 + \frac{2\|(I - Q_tQ_t^\mathsf{T})E\|_{2\to\infty}}{\mathsf{gap}} \cdot \left(1 + \frac{2\|E\|_2}{\lambda_r - \lambda_{r+1}}\right).$$

$$(10)$$

The main drawback of using Equation (10) is that the substitutions used above are not accurate until $Q_tQ_t^\mathsf{T}$ is sufficiently close to $VV^\mathsf{T}$. This is observed empirically in Section 4, as $\mathsf{res}_{2\to\infty}(t)$ is looser than average in the first few iterations.

Another practical concern is evaluating the quality of the bound in Corollary 1; there is no known method for computing the $2 \to \infty$ subspace distance $\min_{Z\in\mathbb{O}_r}\|\hat{V} - VZ\|_{2\to\infty}$ in closed form or via some globally convergent iterative method. However, rather than computing $Z_\star = \operatorname{argmin}_{Z\in\mathbb{O}_r}\|\hat{V} - VZ\|_{2\to\infty}$, we can instead substitute $Z_F = \operatorname{argmin}_{Z\in\mathbb{O}_r}\|\hat{V} - VZ\|_F$, the minimizer of the so-called *orthogonal Procrustes problem*, whose solution can be obtained via the SVD of $V^\mathsf{T}\hat{V}$ [27], as a proxy for tracking the behavior of the $\ell_{2\to\infty}$ distance; this is precisely the solution used by [15] to study perturbations on the $\ell_{2\to\infty}$ distance. Via standard arguments, we are able to show that the aforementioned proxy $\|\hat{V} - VZ_F\|_{2\to\infty}$ enjoys a similar convergence guarantee with an additional multiplicative factor of $\sqrt{r}$, which is typically negligible compared to $n$ – the details are in the supplementary material.

## 4 Applications

In this section, we present a set of numerical experiments illustrating the results of our analysis in practice, as well as the advantages of the proposed stopping criterion. Importantly, in our applications, *entry-wise* error is the natural criterion, often because what matters for the downstream task is an ordering induced by computed eigenvectors. The supplementary material contains more details about the implementation and the experimental setup.

**Synthetic examples.** To verify our theory and get a sense of the tightness of our bounds on convergence rates, we first test on synthetic data. To this end, we generate matrices as follows, given a pair of matrix and subspace dimensions $(n, r)$:

1. Sample a matrix from $\mathbb{O}_n$ uniformly at random (see [39] for details) and select $r$ of its columns uniformly at random to form $V$.
2. generate $\lambda_i \equiv \rho^{i-1}$, for a decay factor $\rho = 0.95$.

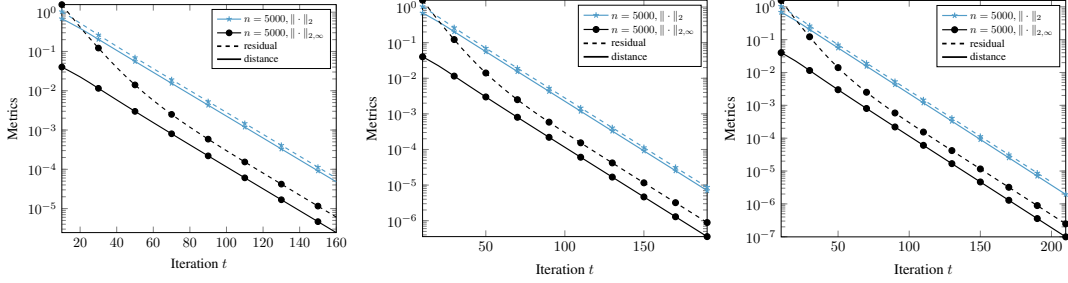

**Figure 1:** Distances (**solid** lines) and residuals (**dashed** lines) for synthetic examples with $r = 50$ and target accuracies $\varepsilon = 10^{-4}$ (**left**), $\varepsilon = 10^{-5}$ (**middle**) and $\varepsilon = 10^{-6}$ (**right**). Each plot corresponds to an independently generated synthetic example.

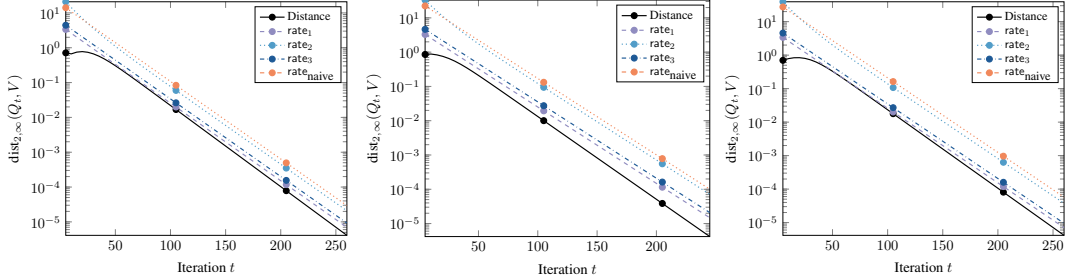

**Figure 2:** Distance (**solid** lines) and convergence rates from Equation (11) for matrix and subspace dimensions $(n, r) = (1000, 10)$ (**left**); $(3500, 15)$ (**middle**); and $(8000, 20)$ (**right**). Our rate$_3$ from Proposition 1 tracks the "idealized" rate rate$_1$ closely in the synthetic data examples.

3. Form $A = [V \;\; V_\perp] \Lambda [V \;\; V_\perp]^{\mathsf{T}}$, where $V_\perp$ is initialized as a random subset of the columns of the identity matrix, and subsequently orthogonalized against $V$.

We compare distances and residuals for synthetic examples with $n = 5000$ and $r = 50$ and various stopping thresholds $\varepsilon$ for the residuals (Figure 1). Each plot in Figure 1 corresponds to a different matrix generated independently according to the aforementioned scheme. While the $\ell_2$ norm residual closely tracks the corresponding distance, the residual from Equation (10) overshoots by a small multiplicative factor, suggesting that the large constants in Proposition 3 may only be necessary in pathological cases and could be reduced in practice. Moreover, the $\ell_{2 \to \infty}$ norm residual can substantially overestimate the actual distance in the first few iterations, as the estimate of Equation (10) depends on $Q_t Q_t^{\mathsf{T}}$ not being "too far" from $VV^{\mathsf{T}}$. The gap narrows after a few dozen iterations.

In addition, we examine the looseness of the bounds from Propositions 1 and 2 for the same experiment (Figure 2). We evaluate the following rates:

$$\text{rate}_1(t) := \left(\frac{\lambda_{r+1}}{\lambda_r}\right)^t \cdot \frac{\text{dist}_{2 \to \infty}(Q_0, V)}{\sqrt{1 - d_0^2}}, \quad \text{rate}_2(t) := \text{ rate from Proposition 2},$$

$$\text{rate}_3(t) := \text{ rate from Proposition 1}, \quad \text{rate}_{\text{naive}}(t) := \left(\frac{\lambda_{r+1}}{\lambda_r}\right)^t \frac{d_0}{\sqrt{1 - d_0^2}} \tag{11}$$

Here, rate$_1$ is an idealized rate that mirrors classical convergence results for the $\ell_2$ norm [24, Theorem 8.2.2]; on the other hand, the naive rate just measures the $\ell_2$ subspace distance. In all the synthetic examples we generated, Assumption 1 was verified to hold with constant $C < 2$ for all elapsed iterations $t$.

Remarkably, for a range of dimensions $n$ and $r$ we find that rate$_3$ (which uses Proposition 1) closely tracks the "idealized" rate$_1$ on these synthetic matrices (Figure 2). Also, rate$_2$ (which uses Proposition 2) is a looser upper bound. This agrees with our theoretical analysis, as $\lambda_{r+2}$ is only moderately smaller than $\lambda_{r+1}$ in our synthetic matrix construction. Finally, as expected, the naive rate is the loosest bound.

**Eigenvector centrality.** Next, we develop an experiment for network centrality, where the task is to measure the influence of nodes in a graph [41]. Each node is assigned a score, which is a function of

**Table 1:** Summary statistics of network datasets.

| Dataset | Citation | # nodes | # edges |
|---|---|---|---|
| CA-HEPPH | [32] | 11204 | 117649 |
| CA-ASTROPH | | 17903 | 197031 |
| GEMSEC-FACEBOOK-ARTIST | [46] | 50515 | 819306 |
| COM-DBLP | [55] | 317080 | 1049866 |
| COM-LIVEJOURNAL | | 3997962 | 34681189 |

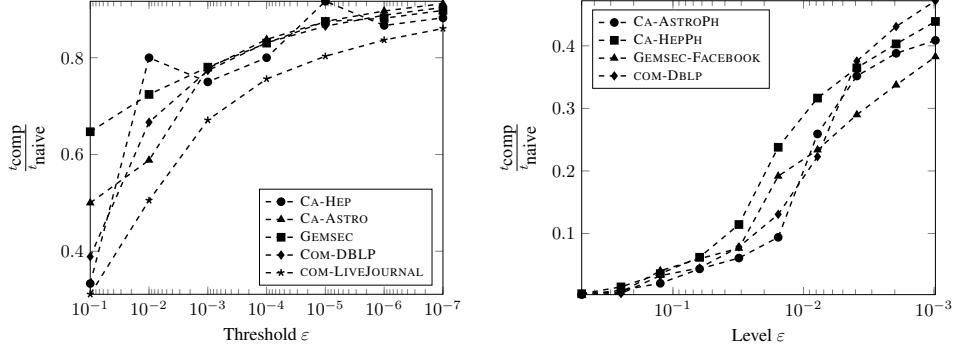

**Figure 3:** Ratio of the number of iterations needed to satisfy the two stopping criteria outlined in (13), for thresholds $\varepsilon = 10^{-k}$, for computing eigenvector centrality to find the $\lfloor\sqrt{n}\rfloor$ most influential nodes (**left**) and computing the leading $r$ eigenvectors for spectral clustering (**right**). Our analysis and stopping criteria enable significantly fewer iterations.

the graph topology, and a typical underlying assumption is that a node with a high score contributes a larger influence to its adjacent ones. Here, we consider *eigenvector centrality*, which is one the standard measures in network science. Given a graph $G = (V, E)$; the eigenvector centrality score of a node $u$, $x_u > 0$, is defined as a solution to the following equation:

$$x_u := \frac{1}{\lambda} \sum_{v \in V} A_{uv} x_v, \ A_{uv} := \begin{cases} 1, & \text{if } u \text{ links to } v \\ 0, & \text{otherwise} \end{cases}, \tag{12}$$

where $\lambda$ is a proportionality constant. Here, node $u$'s scores depend linearly on all of its neighbors' scores. Under the positivity requirement of $x_u$ and provided that the graph is connected and non-bipartite, rearranging and the Perron-Frobenius theorem show that $x = v_1$, the leading eigenvector of $A$ (up to scaling). To determine the most influential nodes, we are typically interested in the induced *ordering* of nodes and not the actual scores themselves. Therefore, the $\ell_{2\to\infty}$ distance, which measures $\|v_1 - \hat{v}_1\|_\infty$, is more appropriate than $\|v_1 - \hat{v}_1\|_2$ as a proxy for the quality of the estimate $\hat{v}_1$. To get a correct ranking result, it suffices to have $\|v_1 - \hat{v}_1\|_\infty < (1/2) \cdot \min_{i,j} |v_i - v_j|$. On the other hand, $\|\hat{v}_1 - v_1\|_2$ does not have an interpretable criterion.

We demonstrate the above principle by comparing two stopping criteria: the criterion from Equation (10) with a specified threshold $\varepsilon$ against the "naive" way of stopping when $\|A\hat{v}_1 - \hat{\lambda}\hat{v}_1\|_2 \leq \hat{\lambda}\varepsilon$, where $\hat{\lambda}$ is the current eigenvalue estimate, via the two following stopping times:

$$\begin{aligned} t_{\text{comp}} &:= \min\{t > 0 \mid \mathsf{res}_{2\to\infty}(t) \leq \varepsilon\} \\ t_{\text{naive}} &:= \min\{t > 0 \mid \|A\hat{V}_{:,j} - \hat{\lambda}_j\hat{V}_{:,j}\| \leq \varepsilon\hat{\lambda}_j, \forall j \in \{1, \ldots, r\}\} \end{aligned}. \tag{13}$$

For a user-specified tolerance $\varepsilon$, we expect that using our $\ell_{2\to\infty}$ error measurements and our corresponding stopping criteria will tell us that we can be confident in our solution much more quickly. This is indeed the case — using our methodology provides a substantial reduction in computation time on a variety of real-world graphs, whose summary statistics are in Table 1. Figure 3 (left) shows the ratio between the two quantities $t_{\text{comp}}$ and $t_{\text{naive}}$, defined as in Equation (13). In the low-to-medium accuracy regimes, using our stopping method results in **at least a 20–40% reduction in the number of iterations needed**. In this regime, the ranking induced by the approximate eigenvector had typically already converged to the "true" ordering obtained by computing the eigenvector to machine precision.

**Spectral clustering in graphs.**    Another downstream task employing invariant subspaces is spectral clustering, which we study here as a way to partition a graph into well-separated "communities" or "clusters." The standard pipeline is to compute the leading $r$-dimensional eigenspace of the normalized adjacency matrix, where $r$ is the desired number of clusters, The resulting eigenvector matrix provides an $r$-dimensional embedding for each node, which is subsequently fed to a point cloud clustering algorithm such as `k-means` [52]. For our experiment, we use the *deterministic* QR-based algorithm from [14] on the same set of real-world graphs that we used for eigenvector centrality.

In this setup, the eigenvectors (more carefully, a rotation of them) are approximate cluster indicators. Indeed, spectral clustering on graphs is often derived from a continuous relaxation of a combinatorial objective based on these indicators [52]. Thus, we are once again interpreting the eigenvectors entry-wise, and $\ell_{2\to\infty}$ error is a more sensible metric than $\ell_2$ error, This fact has been used to analyze random graph models with cluster structure [35].

In the same manner as the eigenvector centrality experiment, we compare the ratio of iteration counts: $t_{\text{comp}}$ over $t_{\text{naive}}$, as defined in Equation (13) (Figure 3, right). In this case, we see even larger savings. For $\varepsilon$ around $10^{-2}$, our stopping criterion results in 70–80% savings in computation time. While this approximation level may seem crude at first, we can measure the performance of the algorithms in terms of the normalized cut metric, for which spectral clustering is a continuous relaxation [52]. We find that by the time we reach residual level $\varepsilon = 10^{-2}$, the cut value found using the approximate subspace is essentially the same as the one using the subspace computed to numerical precision. Further details about the experiment are provided in the supplementary material.

**Spectral bipartitioning and sweep cuts.**    Another spectral method for finding clusters in graphs is spectral bipartitioning, which aims to find a single cluster of nodes $S$ with small conductance $\phi(S)$:

$$\phi(S) := \frac{\sum_{i\in S, j\notin S} A_{ij}}{\min(A(S), A(S^c))}, \ A(S) := \sum_{i\in S}\sum_{j\in V} A_{ij}.$$

The conductance objective is a standard measure for identifying a good cluster of nodes [48, 33]: if $\phi(S)$ is small, there are not many edges leaving $S$ and there are many edges contained in $S$.

Minimizing $\phi(S)$ is NP-hard, but a spectral method using the eigenvector $v_2$ corresponding to the second largest eigenvalue of the normalized adjacency matrix, often called the *Fiedler vector* [21], provides guarantees. To find the a set with small conductance, the method uses the so-called "sweep cut". After scaling $v_2$ by the inverse square root of degrees, we sort the nodes by their value in the eigenvector, and then consider the top-$k$ nodes as a candidate set $S$ for all values of $k$. The value of $k$ that gives the smallest conductance produces a set $S$ satisfying $\phi(S) \le 2\sqrt{\min_{S'}\phi(S')}$, which is the celebrated Cheeger inequality [13].

As in the case of eigenvector centrality, what matters is the *ordering* induced by the eigenvector, making a $\ell_{2\to\infty}$ stopping criterion more appropriate. As a heuristic, one might consider just making $\ell_2$ tolerance larger (by the norm equivalence factor) using a level of $\varepsilon$ for the $\ell_{2\to\infty}$ distance and $\varepsilon \cdot \sqrt{n}$ for the $\ell_2$ distance. However, this can substantially reduce the solution quality. This is illustrated in Figure 4, where we plot the conductance values obtained in the sweep cut as a function of the size of the set on COM-DBLP. This is a sweep cut approximation of a network community profile plot [7, 33], which visualizes cluster structure at different scales. Using the naive $\ell_2$ stopping criterion provides the same solution quality but requires more iterations. In the case of $\varepsilon = 10^{-4}$ in Figure 4, our methods produce 20% computational savings. Finally, the heuristic $\varepsilon \cdot \sqrt{n}$ tolerance for the $\ell_2$ stopping criterion produces a cruder solution and finds a set with larger conductance.

## 5    Conclusions

The broad applicability of spectral methods, coupled with the prevalence of entry-wise / row-wise interpretations of eigenspaces strongly motivates imbuing our computational methods with appropriate stopping criteria. Our theoretical results demonstrate just how much smaller the $\|\cdot\|_{2\to\infty}$ subspace distance can be than traditional measures, an observation supported by experiment. In fact, the accuracy with which we compute eigenvectors can have a non-trivial impact on downstream applications — if we would like use fewer iterations to save time we must do so carefully, and our

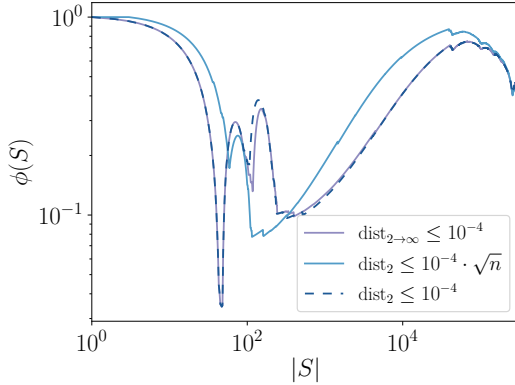

**Figure 4:** Sweep cut profile (cut conductance vs. cardinality) for COM-DBLP. For a fixed $\varepsilon$, our $\ell_{2\to\infty}$ stopping criterion leads to faster convergence. Increasing the tolerance for $\ell_2$ by the norm equivalence factor produces lower-quality solutions. Here $t_{\text{comp}} = 1135$ vs. $t_{\text{naive}} = 1378$ iterations.

new stopping criterion provides an easy to implement way to do this that comes at essentially no cost and with strong guarantees.

From a theoretical perspective, it may seem sufficient to use norm equivalence and simply appeal to spectral norm convergence, which can incur an extra $\mathcal{O}(\log n)$ factor at most when computing subspaces. However, such reasoning only applies to the very-high-accuracy regime. As demonstrated by our experiments, moderate levels of accuracy often suffice for downstream applications, in which case our stopping criterion allows for highly nontrivial computational savings (up to 70% fewer iterations).

## Broader impact

Due to the pervasiveness of spectral methods in machine learning and data mining, our results may be embedded in applications having a wide range of ethical and societal consequences. Indeed, given the fact that eigensolvers are typically used as linear algebra primitives, our work "inherits" the ethical and societal consequences of the context in which its results are applied, as well as the potential implications of "failure" (*e.g.*, if our stopping criterion severely underestimates the true approximation error).

## Acknowledgements & Funding

We would like to thank the anonymous reviewers for their valuable feedback, which helped improve the presentation of this work.

This research was supported by NSF Award DMS-1830274, ARO Award W911NF19-1-0057, and ARO MURI.

## Footnotes

[1]For example, searching for "arpack" [30] (an iterative eigensolver) in the scikit-learn [44] Github repository reveals that several modules depend on it crucially.

[2] Our results hold for the largest magnitude eigenvalues assuming one defines the eigenvalue gap appropriately later. The simplification to the $r$ algebraically largest eigenvalues being the largest in magnitude avoids burdensome notation without losing anything essential.

[3]This includes Algorithm 1 and other common methods such as (block) Lanczos.

[4]As the perturbed matrix is non-normal, an eigengap condition does not suffice to guarantee that $V$ is the leading invariant subspace of the perturbed matrix. To invoke Proposition 3 with the approximate eigenvectors in the place of $Q$, one relies on the fact that $Q$ approaches the leading eigenvector matrix $V$ by convergence theory of subspace iteration. For more details, we refer the reader to the supplementary material.

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
