[Supplementary Material · supplement.pdf]

# Supplementary Material: Entrywise convergence of iterative methods for eigenproblems

**Vasileios Charisopoulos**
Department of Operations Research & Information Engineering
Cornell University
Ithaca, NY 14853
vc333@cornell.edu

**Austin R. Benson**
Department of Computer Science
Cornell University
Ithaca, NY 14853
arb@cs.cornell.edu

**Anil Damle**
Department of Computer Science
Cornell University
Ithaca, NY 14853
damle@cornell.edu

## 1   Additional experimental details

This section documents hyperparameters and other design choices used to run the experiments in Section 4 of the main paper.

### 1.1   Eigenvector centrality

In all eigenvector centrality experiments, we isolate the largest connected component of the input graph and work exclusively within that component. We work with the unnormalized version of the adjacency matrix, since the normalized version admits the vector $d^{1/2} := \mathrm{diag}(\sqrt{d_1}, \ldots, \sqrt{d_n})$, where $d_i$ is the degree of the $i^{\text{th}}$ node, as its principal eigenvector.

We initialize the estimate $\hat{v}_1 := \frac{1}{\sqrt{n}}\mathbf{1}$, the normalized all-ones vector. In the absence of incoherence, we use the expression of Equation (10) in the main text to evaluate the $\ell_{2\to\infty}$ stopping criterion, and set gap $= \lambda_1 - \lambda_2$ using the values returned by Arpack, to ensure a fair comparison. At each step of the iterative method, we multiply with $\pm 1$ accordingly, to ensure that all entries of the approximate eigenvector are positive.

**Ranking distance.**   To measure the "distance" between the approximate ranking produced by our eigenvector estimate, we employ Kendall's $\tau$ criterion Kendall (1948). In particular, we define

$$\mathrm{dist}_\tau(v_1, \hat{v}_1) := \frac{1 - \tau(v_1, \hat{v}_1)}{2} \qquad (1)$$

to compare the rankings induced by $v_1$ and $\hat{v}_1$. It is easy to verify that when the rankings are identical, $\mathrm{dist}_\tau = 0$, and when the rankings are the most dissimilar, $\mathrm{dist}_\tau = 1$, since $\tau(v_1, \hat{v}_1) \in [-1, 1]$.

The convergence plots for 4 datasets, where we depict the "oracle" $\ell_2$ and $\ell_{2\to\infty}$ subspace distances as well as $\mathrm{dist}_\tau$ as a function of the iteration index $t$, are shown in Figure 1. In all cases, we identify the correct ranking when the residual is in the low-to-moderate accuracy regime ($\varepsilon \leq 10^{-4}$).

### 1.2   Spectral clustering

In this section, we describe the methodology used for the spectral clustering experiments in the main text. We opt to use the Algorithm of Damle et al. (2018) which is based on the column-

Figure 1: Distance plots for 4 datasets, for which the top $\lfloor \sqrt{n} \rfloor$ nodes are being ranked. From **left** to **right**: CA-HEPPH, CA-ASTROPH (**top**), GEMSEC, COM-LIVEJOURNAL (**bottom**).

pivoted QR decomposition of an appropriately defined matrix. For completeness, the full algorithm is listed in Algorithm 1. Since the algorithm is deterministic, we do not have to worry about randomness pertaining to initialization (e.g. as in kmeans++), and only run the experiment once for each configuration of parameters.

---

**Algorithm 1** CPQR-based clustering
___

1: **Input**: invariant subspace $V_k \in \mathbb{R}^{n \times r}$
2: Compute the CPQR factorization
$$V_k^\top \Pi = QR,$$
where $\Pi$ is a column selection matrix.
3: Let $\mathcal{C}$ denote the first $k$ columns identified by $\Pi$.
4: Compute the polar factorization
$$(V_k^\top)_{:,\mathcal{C}} = UH.$$
5: **for** $j \in [n]$ **do**
6:     assign node $j$ to cluster
$$C_j := \operatorname*{argmax}_i |(UV_k^\top)_{i,j}|$$
7: **end for**

---

For all the datasets involved, we hand-pick the target number of clusters $r$ by inspecting the successive ratios of the leading few eigenvalues and setting $r$ so that the ratio $\frac{\lambda_{r+1}}{\lambda_r}$ is small, but also taking into account the fact that we don't want $r$ to be too small. Additionally, we use the regularized version of the normalized adjacency matrix $A_\rho$ Amini et al. (2013), which augments the adjacency and degree matrices $A, D$ using a regularization parameter $\rho$:

$$A_\rho := A + \frac{\rho}{n} \mathbf{1}\mathbf{1}^\top, \quad D_\rho := D + \rho \tag{2}$$

Table 1: Parameters for spectral clustering

| Dataset | $r$ | $\tau$ |
|---------|-----|--------|
| CA-HEPPH | 17 | 1.0 |
| CA-ASTROPH | 6 | 1.0 |
| GEMSEC | 12 | 1.0 |
| DBLP | 28 | 5.0 |

Following standard practice Qin & Rohe (2013); Zhang & Rohe (2018), we set $\rho$ equal to a constant which is near the average degree of the graph and then perform the eigendecomposition of

$$\tilde{A}_\rho = D_\rho^{-1/2} A_\rho D_\rho^{-1/2} + I,$$

shifting by $+I$ to ensure that the algebraically largest eigenvalues are also the largest in magnitude, in order for subspace iteration to be applicable. We summarize the hyperparameter choices for each dataset in Table 1. To evaluate the quality of a given clustering assignment, we use the *normalized*

Figure 2: Ratio of iterations required to satisfy $\mathrm{res}_{2\to\infty}(t) \leq \varepsilon$ ($t_{\mathrm{comp}}$) over number of iterations required to satisfy $\mathrm{res}_2(t) \leq \varepsilon$ ($t_{\mathrm{naive}}$) in eigenvector centrality computations.

*cut* metric. Specifically, given a vertex set $V$ and a *partition* $(S, S^c)$ such that $V = S \cup S^c$, we define the conductance of the cut induced by $S$ as

$$\phi(S) := \frac{\sum_{i \in S, j \notin S} A_{ij}}{A(S)} \quad A(S) := \sum_{i \in S} \sum_{j \in V} A_{ij} \tag{3}$$

Note that in (3), $A$ refers to the **unnormalized** adjacency matrix, with $A_{ij} = A_{ji} = 1$ if the edge $(i, j)$ exists in the graph, and 0 otherwise. Then any clustering assignment with $k$ clusters induces $k$ partitions $\{(S_k, S_k^c)\}$, for which the normalized cut metric is defined as

$$\mathrm{ncut}(S_1, \ldots, S_k) := \frac{1}{2} \sum_{i=1}^{k} \phi(S_k). \tag{4}$$

Figure 3 depicts the value of $\mathrm{ncut}(S_1, \ldots, S_k)$ when the input to Algorithm 1 is computed using subspace iteration, using the proposed stopping criterion, for different levels $\varepsilon$. Having established that low-to-moderate accuracy is sufficient for this problem, we plot the ratio of $t_{\mathrm{comp}}$ over $t_{\mathrm{naive}}$; the former is the number of iterations required to satisfy $\mathrm{res}_{2\to\infty}(t) \leq \varepsilon$, while the latter is the number of iterations required to satisfy $\mathrm{res}_2(t) := \left\| A\hat{v}_t - \hat{\lambda}_t \hat{v}_t \right\| \leq \hat{\lambda}_t \varepsilon$. We observe computational gains of over $50\%$ in all cases.

Figure 3: Value of ncut$(S_1, \ldots, S_k)$ for various datasets, with $\hat{V}_k$ computed using subspace iteration until the residual drops below level $\varepsilon$, for different values of $\varepsilon$. In all cases, the metric stabilizes while in the low accuracy regime ($\varepsilon \approx 10^{-2}$). Dashed lines indicate the value of ncut$(S_1, \ldots, S_k)$ found by computing the subspace to machine accuracy.

Figure 4: Ratio of iterations required to satisfy $\mathrm{res}_{2\to\infty}(t) \leq \varepsilon$ ($t_{\mathrm{comp}}$) over number of iterations required to satisfy $\mathrm{res}_2(t) \leq \varepsilon$ ($t_{\mathrm{naive}}$) in the spectral clustering setting, showing computational gains of over 50%.

### 1.3 Empirically verifying Assumption 1

We verified that Assumption 1 from the main text holds in practice for the real world datasets used in the experimental section. Recall that Assumption 1 asks that the matrix $A = V\Lambda V^\mathsf{T} + V_\perp \Lambda_\perp V_\perp^\mathsf{T}$ (where $V \in \mathbb{O}_{n,r}$ is the leading $r$-dimensional invariant subspace of $A$) satisfies:

$$\|V_\perp \Lambda_\perp^t V_\perp^\mathsf{T}\|_\infty \leq C \cdot \lambda_{\max}^t(\Lambda_\perp) \cdot \|V_\perp V_\perp^\mathsf{T}\|_\infty, \tag{5}$$

where $C$ is a constant independent of $n$, for all $t \in \mathbb{N}$. First, observe that for our purposes, we only want this assumption to hold for all $t$ until our iterative algorithm stops. Since all our experiments take fewer than $T = 1500$ iterations to run, we opt to verify (5) for $t \in \{1, \ldots, T\}$. We first rephrase the assumption as

$$\|A^t - V\Lambda^t V^\mathsf{T}\|_\infty \leq C \cdot \lambda_{\max}^t(\Lambda_\perp) \cdot \|I - VV^\mathsf{T}\|_\infty, \tag{6}$$

which can be checked after computing the top $r + 1$ eigenvectors and eigenvalues of $A$; these were computed to machine precision using `eigs`. For $t = 1$ up to $t = T$, we checked (6) exhaustively, and output

$$C := \sup_{t \in \{1,\ldots,T\}} \left\{ \frac{\|A^t - V\Lambda^t V^\mathsf{T}\|_\infty}{\lambda_{r+1}^t \|I - VV^\mathsf{T}\|_\infty} \right\}$$

In all cases, we end up with a constant $C \leq 1.5$.

## 2 Auxiliary results

**Lemma 1** (Incoherence). *Consider a subspace $\mathcal{V}$ of dimension $r$ and a matrix $V \in \mathbb{O}_{n,r}$ whose columns span $\mathcal{V}$. If $\mu$ is the coherence of $V$, i.e. $\|V\|_{2\to\infty} \leq \mu\sqrt{\frac{r}{n}}$, then for its complementary subspace $\mathcal{V}_\perp$ it holds that*

$$\left\|V_\perp V_\perp^\mathsf{T}\right\|_\infty \leq (1 + \mu\sqrt{r}).$$

*Proof.* Observe that $\|A\|_\infty \leq \sqrt{n}\,\|A\|_{2\to\infty}$, tus

$$\left\|V_\perp V_\perp^\mathsf{T}\right\|_\infty = \|I - VV^\mathsf{T}\|_\infty \leq 1 + \|VV^\mathsf{T}\|_\infty$$
$$\leq 1 + \sqrt{n}\,\|VV^\mathsf{T}\|_{2\to\infty} \leq 1 + \sqrt{n}\mu\sqrt{r/n}.$$

$\square$

The next theorem, originally stated without assuming symmetry, is adapted for the case of a symmetric initial matrix.

**Theorem 1** (Theorem 5.1 in (Damle & Sun, 2020)). *Suppose $\tilde{A} = A + E$ with $A$ symmetric, having eigenvalue decomposition $A = V\Lambda V^\mathsf{T} + V_\perp \Lambda_\perp V_\perp^\mathsf{T}$, where $V \in \mathbb{R}^{n\times r}, V_\perp \in \mathbb{R}^{n\times(n-r)}$ have orthonormal columns. Moreover, let $\mathsf{gap} := \min\left\{\lambda_r - \lambda_{r+1}, \mathsf{sep}_{(2,\infty),V_\perp}(\Lambda, V_\perp \Lambda_\perp V_\perp^\mathsf{T})\right\}$. If $\|E\|_2 \leq \frac{\mathsf{gap}}{5}$, then the leading invariant subspace of $\tilde{A}$, $\tilde{V}$, satisfies*

$$\inf_{O\in\mathbb{O}_r} \left\|\tilde{V} - VO\right\|_{2\to\infty} \leq 8\|V\|_{2\to\infty}\left(\frac{\|E\|_2}{\lambda_r - \lambda_{r+1}}\right)^2 + \frac{2\left\|V_\perp V_\perp^\mathsf{T}EV\right\|_{2\to\infty}}{\mathsf{gap}}$$
$$+ \frac{4\left\|V_\perp V_\perp^\mathsf{T}E\right\|_{2\to\infty}\|E\|_2}{\mathsf{gap}\cdot(\lambda_r - \lambda_{r+1})}. \tag{7}$$

**Lemma 2** (Cape et al. (2019)). *We have*

$$\|AB\|_{2\to\infty} \leq \|A\|_{2\to\infty}\|B\|_2 \tag{8}$$
$$\|AB\|_{2\to\infty} \leq \|A\|_\infty\|B\|_{2\to\infty} \tag{9}$$

*Moreover, for any matrix $V$ with orthonormal columns, it holds that*

$$\|AV^\mathsf{T}\|_{2\to\infty} = \|A\|_{2\to\infty}. \tag{10}$$

We also prove the following claim, which is used throughout the proof of Proposition 1 in the next section.

**Lemma 3.** *We have $\inf_{Z\in\mathbb{O}_r}\left\|\tilde{V} - VZ\right\|_2 \leq \sqrt{2}\mathrm{dist}_2(V, \tilde{V})$.*

*Proof.* Recall the solution of the orthogonal Procrustes problem, given by the SVD of $V^\mathsf{T}\tilde{V}, U\Sigma W^\mathsf{T}$. Since $UW^\mathsf{T} \in \mathbb{O}_r$, with $U^\mathsf{T}U = UU^\mathsf{T} = W^\mathsf{T}W = WW^\mathsf{T} = I_r$, we have

$$\inf_{Z\in\mathbb{O}_r}\left\|\tilde{V} - VZ\right\|_2 \leq \left\|\tilde{V} - VUW^\mathsf{T}\right\|_2 = \sqrt{\sup_x \left\langle x, (\tilde{V} - VUW^\mathsf{T})^\mathsf{T}(\tilde{V} - VUW^\mathsf{T})x\right\rangle} \tag{11}$$

$$= \sqrt{\sup_x \left\langle x, (I - \tilde{V}^\mathsf{T}VUW^\mathsf{T} - WU^\mathsf{T}V^\mathsf{T}\tilde{V} + I)x\right\rangle} \tag{12}$$

$$\overset{(\sharp)}{=} \sqrt{\sup_x \left\langle x, 2(I - W\Sigma W^\mathsf{T})x\right\rangle} = \sqrt{2\left\|I - W\Sigma W^\mathsf{T}\right\|_2} \tag{13}$$

$$= \sqrt{2}\sqrt{\|I - \Sigma\|_2} = \sqrt{2}\sqrt{1 - \sigma_r(V^\mathsf{T}\tilde{V})} \tag{14}$$

$$\overset{(\natural)}{\leq} \sqrt{2}\sqrt{1 - \sigma_r^2(V^\mathsf{T}\tilde{V})} = \sqrt{2}\left\|V^\mathsf{T}\tilde{V}\right\|_2, \tag{15}$$

where $(\sharp)$ follows after replacing $V^\mathsf{T}\tilde{V} = U\Sigma W^\mathsf{T}$ in the expression and gathering terms, while $(\natural)$ simply uses the fact that $\sigma_r(V^\mathsf{T}\tilde{V}) \leq 1$ to upper bound the expression inside the square root. Finally, we use the fact that:
$$1 - \sigma_{\min}^2(V^\mathsf{T}\tilde{V}) = \left\|V_\perp^\mathsf{T}\tilde{V}\right\|_2^2 = \mathrm{dist}_2^2(V, \tilde{V}).$$

$\square$

## 3 Omitted proofs

### 3.1 Proof of Proposition 1

Starting with the definition of the $2 \to \infty$ distance, we have

$$\mathrm{dist}_{2\to\infty}(Q_t, V) = \inf_{Z\in\mathbb{O}_r}\|Q_t - VZ\|_{2\to\infty} = \inf_{Z\in\mathbb{O}_r}\|(VV^\mathsf{T} + V_\perp V_\perp^\mathsf{T})(Q_t - VZ)\|_{2\to\infty} \tag{16}$$

$$\overset{(\sharp)}{\leq} \sqrt{2}\|VV^\mathsf{T}\|_{2\to\infty}\mathrm{dist}_2(Q_t, V) + \|V_\perp V_\perp^\mathsf{T}(Q_t - VZ)\|_{2\to\infty} \tag{17}$$

where ($\sharp$) follows from Lemma 2 and the fact that $\inf_{Z\in\mathbb{O}_r}\|Q_t - VZ\|_2 \leq \sqrt{2}\mathrm{dist}_2(Q_t,V)$. At this point, note that standard convergence results (Saad, 2011; Golub & Van Loan, 2013) state that

$$\mathrm{dist}_2(Q_t,V) \leq \left(\frac{\lambda_{r+1}}{\lambda_r}\right)^t \frac{d_0}{\sqrt{1-d_0^2}},$$

and additionally $\|VV^\mathsf{T}\|_{2\to\infty} \leq \mu\sqrt{\frac{r}{n}}$, where $\mu$ is the coherence of $V$.

For the remainder, let us first recall a fact from the analysis of subspace iteration; the $t^{\text{th}}$ iterate $Q_t$ satisfies

$$Q_t R_t = A^t V^{(0)}, \quad \text{with } R_t \text{ invertible} \quad \Rightarrow \quad V_\perp^\mathsf{T} Q_t = V_\perp^\mathsf{T} A^t V^{(0)} R_t^{-1} = \Lambda_\perp^t V_\perp^\mathsf{T} V^{(0)} R_t^{-1}. \quad (18)$$

Then, notice that $V_\perp^\mathsf{T} V = 0$ and therefore we can rewrite the second term in (22) as

$$\|V_\perp V_\perp^\mathsf{T} Q_t\|_{2\to\infty} \overset{(*)}{=} \|V_\perp \Lambda_\perp^t V_\perp^\mathsf{T} Q_0 R_t^{-1}\|_{2\to\infty} \overset{(\flat)}{=} \inf_{Z\in\mathbb{O}_r} \|V_\perp \Lambda_\perp^t V_\perp^\mathsf{T}(Q_0 - VZ) R_t^{-1}\|_{2\to\infty} \quad (19)$$

$$\overset{(\natural)}{\leq} \inf_{Z\in\mathbb{O}_r} C \|V_\perp V_\perp^\mathsf{T}\|_\infty \lambda_{r+1}^t \|(Q_0 - VZ) R_t^{-1}\|_{2\to\infty} \quad (20)$$

$$\leq C \|V_\perp V_\perp^\mathsf{T}\|_\infty \lambda_{r+1}^t \underbrace{\inf_{Z\in\mathbb{O}_r} \|Q_0 - VZ\|_{2\to\infty}}_{=\mathrm{dist}_{2\to\infty}(Q_0,V)} \|R_t^{-1}\|_2 \quad (21)$$

where ($*$) follows from Eq. (18), ($\flat$) holds since we can reintroduce $VZ$ for any $Z$, as $V_\perp^\mathsf{T} V = 0$, ($\natural$) holds after combining Eq. (9) and Assumption 1 from the main text, and the last inequality is Eq. (8). Notice that $\|R_t^{-1}\|_2 = \frac{1}{\sqrt{1-d_0^2}}\lambda_r^{-t}$, by tracing the proof of (Golub & Van Loan, 2013, Theorem 8.2.2). Finally, by Lemma 1, $\|V_\perp V_\perp^\mathsf{T}\|_\infty \leq 1 + \mu\sqrt{r}$. □

### 3.2 Proof of Proposition 2

For simplicity, let us define $\tilde{V} := [V \quad v_{r+1}] \in \mathbb{R}^{n\times(r+1)}$ and $\tilde{V}_\perp$ for the remaining $n-r-1$ eigenvectors forming a basis of $\mathbb{R}^n$. Similarly, let $\tilde{\Lambda}_\perp = \mathrm{diag}(\lambda_{r+2},\ldots,\lambda_n)$. Starting with the definition of the $2\to\infty$ distance, we have

$$\mathrm{dist}_{2\to\infty}(Q_t,V) = \inf_{Z\in\mathbb{O}_r} \|Q_t - VZ\|_{2\to\infty} = \inf_{Z\in\mathbb{O}_r} \|(VV^\mathsf{T} + V_\perp V_\perp^\mathsf{T})(Q_t - VZ)\|_{2\to\infty}$$
$$\overset{(\sharp)}{\leq} \sqrt{2}\|VV^\mathsf{T}\|_{2\to\infty}\,\mathrm{dist}_2(Q_t,V) + \|V_\perp V_\perp^\mathsf{T}(Q_t - VZ)\|_{2\to\infty} \quad , \quad (22)$$

where ($\sharp$) follows from Lemma 2 in the main text and the fact that $\inf_{Z\in\mathbb{O}_r}\|Q_t - VZ\|_2 \leq \sqrt{2}\mathrm{dist}_2(Q_t,V)$. Now we may rewrite the second term as

$$\|(v_{r+1}v_{r+1}^\mathsf{T} + \tilde{V}_\perp \tilde{V}_\perp^\mathsf{T})Q_t\|_{2\to\infty} \leq \|v_{r+1}v_{r+1}^\mathsf{T} Q_t\|_{2\to\infty} + \|\tilde{V}_\perp \tilde{V}_\perp^\mathsf{T} Q_t\|_{2\to\infty}$$
$$= \|v_{r+1}\lambda_{r+1}^t v_{r+1}^\mathsf{T} Q_0 R_t^{-1}\|_{2\to\infty} + \|\tilde{V}_\perp \tilde{\Lambda}_\perp^t \tilde{V}_\perp^\mathsf{T} Q_0 R_t^{-1}\|_{2\to\infty}. \quad (23)$$

Pulling $\lambda_{r+1}^t$ out of the first norm in (23) yields

$$\|v_{r+1}v_{r+1}^\mathsf{T}(Q_0 - VZ_\star)\|_{2\to\infty}\|R_t^{-1}\|_2 \leq \|v_{r+1}v_{r+1}^\mathsf{T}\|_\infty\,\mathrm{dist}_{2\to\infty}(Q_0,V)\cdot\frac{\lambda_r^{-t}}{\sqrt{1-d_0^2}},$$

after using Lemma 2 and the fact that $\|R_t^{-1}\|_2 \leq \frac{\lambda_r^{-t}}{\sqrt{1-d_0^2}}$, while the second norm in (23) can be upper bounded by

$$\left\|\tilde{V}_\perp\tilde{\Lambda}_\perp^t\right\|_2 \|\tilde{V}_\perp^\mathsf{T} Q_-\|_2 \|R_t^{-1}\|_2 = \left(\frac{\lambda_{r+2}}{\lambda_r}\right)^t \frac{\mathrm{dist}_2(Q_0,\tilde{V})}{\sqrt{1-d_0^2}},$$

but as the respective subspaces satisfy $\mathcal{V}\subset\tilde{\mathcal{V}}$ we have $\mathrm{dist}_2(Q_0,\tilde{V}) \leq \mathrm{dist}_2(Q_0,V)$. Combining all the ingredients above completes the proof. □

## 3.3 Proof of Proposition 3

The condition on $\|E\|_2$ combined with the assumption that $Q$ is the leading invariant subspace of the perturbed matrix $A - EQ^\mathsf{T}$ allows us to apply Theorem 1 for the perturbation $EQ^\mathsf{T}$, from which we deduce that the approximate eigenvector matrix $V$ satisfies

$$\mathrm{dist}_{2\to\infty}(Q, V) \le 8 \|V\|_{2\to\infty} \left( \frac{\|E\|_2}{\lambda_r - \lambda_{r+1}} \right)^2 + 2 \frac{\|V_\perp V_\perp^\mathsf{T} EQ^\mathsf{T} V\|_{2\to\infty}}{\mathsf{gap}} + 4 \frac{\|V_\perp V_\perp^\mathsf{T} E\|_{2\to\infty} \|E\|_2}{\mathsf{gap} \cdot (\lambda_r - \lambda_{r+1})}$$

with the appropriate definition of gap. Using Lemma 2, we can upper bound the terms above as

$$\|V_\perp V_\perp^\mathsf{T} EQ^\mathsf{T} V\|_{2\to\infty} \le \|V_\perp V_\perp^\mathsf{T}\|_\infty \|EQ^\mathsf{T} V\|_{2\to\infty} \le \|V_\perp V_\perp^\mathsf{T}\|_\infty \|E\|_{2\to\infty} \underbrace{\|Q^\mathsf{T} V\|_2}_{\le 1}, \qquad (24)$$

and similarly for the term $\|V_\perp V_\perp^\mathsf{T} E\|_{2\to\infty}$. $\qquad\square$

# 4 Miscellanea

**Discussion: eigenvalue localization issues.** We briefly address the issue of when we can safely assume that the approximate invariant subspace $Q$, utilized in Proposition 3, is the **leading** invariant subspace of the perturbed matrix $A - EQ^\mathsf{T}$. While the matrix of Ritz values, $S$, is within $\sqrt{2}\|E\|_2$ distance of a set of $r$ eigenvalues of $A$, we do not know whether or not these eigenvalues correspond to the largest (in magnitude) eigenvalues of $A - EQ^\mathsf{T}$.

In this case, one has to appeal to algorithm-specific arguments. Recall that $A$ has spectral decomposition $A = V\Lambda V^\mathsf{T} + V_\perp \Lambda_\perp V_\perp^\mathsf{T}$, where $\Lambda$ contains the dominant $r$ eigenvalues. Let $Q_\perp \in \mathbb{O}_{n,n-r}$ be orthogonal to the approximate eigenvector matrix $Q \in \mathbb{O}_{n,r}$. Then the following

$$\begin{bmatrix} Q^\mathsf{T} \\ Q_\perp^\mathsf{T} \end{bmatrix} (A - EQ^\mathsf{T}) [Q \quad Q_\perp] = \begin{bmatrix} S & Q^\mathsf{T}(A - EQ^\mathsf{T})Q_\perp \\ Q_\perp^\mathsf{T} QS & Q_\perp^\mathsf{T}(A - EQ^\mathsf{T})Q_\perp \end{bmatrix} = \begin{bmatrix} S & Q^\mathsf{T} AQ_\perp \\ \mathbf{0} & Q_\perp^\mathsf{T} AQ_\perp \end{bmatrix}$$

is a Schur decomposition of $A - EQ^\mathsf{T}$, with its eigenvalues being the union $S \cup \Lambda(Q_\perp^\mathsf{T} AQ_\perp)$ – the objective becomes showing that $\|\Lambda(Q_\perp^\mathsf{T} AQ_\perp)\|_2$ is sufficiently small, after enough progress of the algorithm. By the variational characterization of singular values for symmetric matrices, we have

$$\|Q_\perp^\mathsf{T} AQ_\perp\|_2 = \sup_{x \in \mathbb{S}^{n-1}} |\langle x, Q_\perp^\mathsf{T} AQ_\perp x \rangle| \tag{25}$$

$$= \sup_{x \in \mathbb{S}^{n-1}} |\langle x, Q_\perp^\mathsf{T} V\Lambda V^\mathsf{T} Q_\perp x \rangle + \langle x, Q_\perp^\mathsf{T} V_\perp \Lambda_\perp V_\perp^\mathsf{T} Q_\perp x \rangle| \tag{26}$$

$$\overset{(*)}{\le} |\lambda_1(A)| \|Q_\perp^\mathsf{T} V\|_2^2 + |\lambda_{r+1}(A)| \underbrace{\|Q_\perp^\mathsf{T} V_\perp\|}_{\le 1} \tag{27}$$

Therefore, as soon as $\mathrm{dist}_2(V, Q) \le \sqrt{\varepsilon}$, we know that $\Lambda(Q_\perp^\mathsf{T} AQ_\perp) \le |\lambda_1|\varepsilon + |\lambda_{r+1}|$; thus when both $\|E\|_2$ and $\varepsilon$ are small enough, we can "match" $S$ with the leading invariant subspace of $A - EQ^\mathsf{T}$, via the leading eigenvalues of $A$ itself.

**Discussion: entrywise convergence of Procrustes solution.** Let $V_1, \hat{V}_1$ be a pair of matrices with orthogonal columns. Recall that the Procrustes solution is the solution to the following matrix nearness problem:

$$Z_F := \operatorname*{argmin}_{Z \in \mathbb{O}_r} \left\| \hat{V}_1 Z - V_1 \right\|_F, \tag{28}$$

for which the solution is available via the SVD of $\hat{V}_1^\mathsf{T} V_1$ Higham (1988). For the iterates $\{Q_t\}_{t \in \mathbb{N}}$ produced by Algorithm 1 in the main text, notice that

$$\inf_{Z \in \mathbb{O}_r} \|Q_t - VZ\|_{2\to\infty} \le \|Q_t - VZ_F\|_{2\to\infty} \le \mu\sqrt{\frac{r}{n}} \|Q_t - VZ_F\|_2 + \|V_\perp V_\perp^\mathsf{T} Q_t\|_{2\to\infty}. \tag{29}$$

For the first term, using the definition of $Z_F$ and choosing $Z_2 := \operatorname{argmin}_{Z \in \mathbb{O}_r} \|Q_t - VZ\|_2$, we may obtain

$$\|Q_t - VZ_F\|_2 \le \|Q_t - VZ_F\|_F \le \|Q_t - VZ_2\|_F \tag{30}$$

$$\overset{(\sharp)}{\le} \sqrt{2r} \cdot \|Q_t - VZ_2\|_2 \overset{(\flat)}{\le} 2\sqrt{r} \cdot \mathrm{dist}_2(Q_t, V), \tag{31}$$

where ($\sharp$) follows by the fact that $\operatorname{rank}(Q_t - VZ_2) \leq 2r$ combined with norm equivalence, and ($\flat$) follows from Lemma 3. Together with the second term in Eq. (29), these can be analyzed as in the proofs of Propositions 1 and 2.

**Discussion: preliminary convergence results *without* Assumption 1.**   Here, we provide a proof showing that the convergence of subspace iteration w.r.t. the $2 \to \infty$ norm improves upon the spectral norm results without the need for Assumption 1 from the main text on the data matrix's eigenspaces. We show this by studying a "worse-case" version of $A$, $\tilde{A}$, instead; given $A = V\Lambda V^{\mathsf{T}} + V_\perp \Lambda_\perp V_\perp^{\mathsf{T}}$, we define $\tilde{A}$ as

$$\tilde{A} := V\Lambda V^{\mathsf{T}} + \lambda_{r+2}(A) \cdot V_\perp V_\perp{}^{\mathsf{T}}. \tag{32}$$

In the forthcoming proof, we denote $\tilde{\Lambda}_\perp := \lambda_{r+2}(A) \cdot I_{n-r}$. The Proposition below gives an improved rate compared to the analysis w.r.t. spectral norm convergence, albeit for a limited set of spectra.

**Proposition.** *The iterates $\{Q_t\}_{t \in [T]}$ produced by Algorithm 1 in the main text with initial guess $V^{(0)}$ satisfy*

$$\operatorname{dist}_{2 \to \infty}(Q_t, V) \leq 3\frac{1 + \mu\sqrt{r}}{\sqrt{1 - d_0^2}} \left(\frac{\lambda_{r+2}}{\lambda_r}\right)^t \cdot \operatorname{dist}_{2 \to \infty}(V^{(0)}, V)$$

$$+ \mu\sqrt{\frac{r}{n}}\left(\frac{\lambda_{r+1}}{\lambda_r}\right)^t \cdot \tan(\theta_0) + \max\left\{\frac{\lambda_{r+1}^t - \lambda_{r+2}^t}{\lambda_r^t}, \frac{\lambda_{r+2}^t - \lambda_n^t}{\lambda_r^t}\right\} \cdot \tan(\theta_0), \tag{33}$$

*where $\tan(\theta_0) := \frac{d_0}{\sqrt{1-d_0^2}}$, $d_0 := \operatorname{dist}_2(Q_0, V)$.*

*Proof.* Let us introduce some notation to be used in the proof; given the true subspace $Q$, we write $\operatorname{dist}_{\|\cdot\|, \perp}(A, B) := \operatorname{dist}_{\|\cdot\|}(V_\perp V_\perp^{\mathsf{T}} A, V_\perp V_\perp^{\mathsf{T}} B)$. By splitting up $Q_t$ into its projections to $V$ and $V_\perp$ respectively, we can upper bound the desired distance in the following way:

$$\operatorname{dist}_{2,\infty}(Q_t, V) = \inf_Z \|Q_t - VZ\|_{2 \to \infty} = \inf_Z \|(VV^{\mathsf{T}} + V_\perp V_\perp^{\mathsf{T}})Q_t - VZ\|_{2 \to \infty}$$

$$\leq \inf_Z \|VV^{\mathsf{T}}(Q_t - VZ)\|_{2 \to \infty} + \|V_\perp V_\perp^{\mathsf{T}} Q_t\|_{2 \to \infty} \tag{34}$$

$$\leq \|V\|_{2 \to \infty} \operatorname{dist}_2(Q_t, V) + \operatorname{dist}_{2 \to \infty, \perp}(Q_t, V) \tag{35}$$

since $V_\perp^{\mathsf{T}} V = 0$. The first term in (35) is upper bounded by

$$\mu\sqrt{\frac{r}{n}}\left(\frac{\lambda_{r+1}}{\lambda_r}\right)^t \tan(\theta_0),$$

(where $\mu$ is the coherence of $V$), which is known from the standard convergence analysis of Algorithm 1 measured in the spectral norm. In addition, using the triangle inequality for the second term in (35), we can further upper bound

$$\|V_\perp V_\perp^{\mathsf{T}} Q_t\|_{2 \to \infty} = \operatorname{dist}_{2 \to \infty, \perp}(Q_t, V) \leq \operatorname{dist}_{2 \to \infty, \perp}(Q_t, \tilde{Q}_t) + \operatorname{dist}_{2 \to \infty, \perp}(\tilde{Q}_t, V) \tag{36}$$

where $\tilde{Q}_t$ is the aforementioned "ghost" iterate resulting from applying Algorithm 1 to the matrix $\tilde{A}$, which is defined as

$$\tilde{A} := [V \quad V_\perp] \begin{bmatrix} \Lambda & 0 \\ 0 & \lambda_{r+2}(A)I_{n-r} \end{bmatrix} [V \quad V_\perp]^{\mathsf{T}}, \tag{37}$$

and obviously $Q_0 = \tilde{Q}_0 := V^{(0)}$. In the forthcoming steps, we bound each distance above separately. For the second term in (36), we have:

**Lemma 4.** *The iterates $\{\tilde{Q}_t\}_{t \in [T]}$ produced by Algorithm 1 when applied to $\tilde{A}$, as defined in (37), satisfy*

$$\operatorname{dist}_{2 \to \infty, \perp}(\tilde{Q}_t, V) \leq \left(\frac{\lambda_{r+2}}{\lambda_r}\right)^t \frac{1}{\sqrt{1 - d_0^2}}(1 + \mu\sqrt{r}) \cdot \operatorname{dist}_{2 \to \infty}(Q_0, V) \tag{38}$$

*where $d_0 := \operatorname{dist}_2(Q_0, V)$, and $\mu$ is the coherence of $V$.*

*Proof.* We build heavily on the proof of the analogous convergence result for the spectral norm given in (Golub & Van Loan, 2013). First, observe that $\tilde{A}$ has the same eigenvectors as $A$ and same first $r$ as well as last $n - r - 1$ eigenvalues.

From the proof of (Golub & Van Loan, 2013, Theorem 8.2.2), we know that $\tilde{Q}_t \tilde{R}_t = \tilde{A}^t V^{(0)}$, with $\tilde{R}_t$ invertible and satisfying

$$\left\| \tilde{R}_t^{-1} \right\|_2 \leq \frac{\lambda_r^{-t}}{\sqrt{1 - d_0^2}}, \quad d_0 := \mathrm{dist}_2(V^{(0)}, V), \tag{39}$$

Then we have

$$
\begin{aligned}
\mathrm{dist}_{2\to\infty,\perp}(\tilde{Q}_t, V) = \left\| V_\perp V_\perp^\mathsf{T} \tilde{Q}_t \right\|_{2\to\infty} &= \left\| V_\perp \tilde{\Lambda}_\perp^\mathsf{T} V_\perp^\mathsf{T} V^{(0)} \tilde{R}_t^{-1} \right\|_{2\to\infty} \\
&\leq \inf_Z \left\| V_\perp \tilde{\Lambda}_\perp^\mathsf{T} V_\perp^\mathsf{T} (V^{(0)} - VZ) \right\|_{2\to\infty} \left\| \tilde{R}_t^{-1} \right\|_2 \\
&\overset{(\sharp)}{\leq} \left( \frac{\lambda_{r+2}}{\lambda_r} \right)^t \left\| V_\perp V_\perp^\mathsf{T} \right\|_\infty \mathrm{dist}_{2\to\infty}(V^{(0)}, V) \frac{1}{\sqrt{1 - d_0^2}}
\end{aligned}
\tag{40}
$$

where the second step of the proof uses (8) and ($\sharp$) uses Equations (9) and (39). Finally, an appeal to Lemma 1 yields the desired expression. □

For the first term in (36), we follow a similar approach and write (for $Z$ attaining the infimum in the definition of the subspace distance):

$$\mathrm{dist}_{2\to\infty,\perp}(Q_t, \tilde{Q}_t) = \left\| V_\perp V_\perp^\mathsf{T}(Q_t - \tilde{Q}_t Z) \right\|_{2\to\infty} = \left\| V_\perp \Lambda_\perp^t V_\perp^\mathsf{T} R_t^{-1} - V_\perp \tilde{\Lambda}_\perp^t V_\perp^\mathsf{T} \tilde{R}_t^{-1} Z \right\|_{2\to\infty},$$

where again we recall from the proof of (Golub & Van Loan, 2013, Theorem 8.2.2) that

$$V_\perp^\mathsf{T} Q_t = \Lambda_\perp^t V_\perp^\mathsf{T} V^{(0)} R_t^{-1}, \ V_\perp^\mathsf{T} \tilde{Q}_t = \lambda_{r+2}^t V_\perp^\mathsf{T} V^{(0)} \tilde{R}_t^{-1},$$

as in the proof of Lemma 4. Consequently, we can write $\tilde{R}_t^{-1} Z = R_t^{-1} + (\tilde{R}_t^{-1} Z - R_t^{-1})$ and substitute above to obtain

$$
\begin{aligned}
\left\| V_\perp V_\perp^\mathsf{T}(Q_t - \tilde{Q}_t Z) \right\|_{2\to\infty} &\leq \left\| V_\perp \left( \Lambda_\perp^t - \lambda_{r+2}^t I_{n-r} \right) V_\perp^\mathsf{T} V^{(0)} R_t^{-1} \right\|_{2\to\infty} \\
&\quad + \lambda_{r+2}^t \left\| V_\perp V_\perp^\mathsf{T} V^{(0)} (\tilde{R}_t^{-1} Z - R_t^{-1}) \right\|_{2\to\infty},
\end{aligned}
\tag{41}
$$

after appropriate rearrangements and the triangle inequality. Rewriting $V_\perp^\mathsf{T} V^{(0)} = V_\perp^\mathsf{T}(V^{(0)} - VZ_{2\to\infty})$ yields

$$
\begin{aligned}
\left\| V_\perp V_\perp^\mathsf{T} V^{(0)} (\tilde{R}_t^{-1} - R_t^{-1}) \right\|_{2\to\infty} &\leq \left\| V_\perp V_\perp^\mathsf{T} \right\|_\infty \mathrm{dist}_{2\to\infty}(V^{(0)}, V) \left( \left\| \tilde{R}_t^{-1} \right\|_2 + \left\| R_t^{-1} \right\|_2 \right) \\
&\leq 2 \left( 1 + \mu \sqrt{r} \right) \frac{1}{\lambda_r^t} \frac{\mathrm{dist}_{2,\infty}(V^{(0)}, V)}{\sqrt{1 - d_0^2}},
\end{aligned}
\tag{42}
$$

since $R_t^{-1}$ and $\tilde{R}_t^{-1}$ both satisfy (39) as $A$ and $\tilde{A}$ have the same first $r + 1$ eigenvalues. Finally

$$\left\| \Lambda_\perp^t - \lambda_{r+2}^t I_{n-r} \right\|_2 = \max \left\{ |\lambda_{r+1}^t - \lambda_{r+2}^t|, |\lambda_{r+2}^t - \lambda_n^t| \right\}, \tag{43}$$

which we may use to bound the first term in (41) by noticing

$$\left\| V_\perp (\Lambda_\perp^t - \lambda_{r+2}^t I) V_\perp^\mathsf{T} V^{(0)} R_t^{-1} \right\|_{2\to\infty} \leq \left\| V_\perp \right\|_{2\to\infty} \left\| \Lambda_\perp^t - \lambda_{r+2}^t I \right\|_2 \left\| V_\perp^\mathsf{T} V^{(0)} \right\|_2 \left\| R_t^{-1} \right\|_2 \tag{44}$$

The proof follows by combining Equations (42) to (44), the fact that $d_0 = \left\| V_\perp^\mathsf{T} V^{(0)} \right\|_2$, and Lemma 4. □

## 5  Reproducibility

We provide an open-source implementation of the algorithms and all experiments in `Julia` in the following repository: `https://github.com/VHarisop/entrywise-convergence`. The experiments were run in a machine running Manjaro Linux with 16 GB of RAM and Intel®Core™ i7-7700 CPU @ 3.60 GHz, using Julia version 1.1.