[Reviews · NeurIPS 2020]

Review 1

Summary and Contributions: This paper discusses bounding the errors in approximating eigenvectors when using subspace iteration (a classical generalisation of power iteration). Traditional analysis bounds the l2 subspace distance (dist_2 in the paper). The paper observes that in many ML applications, an entrywise bound is more appropriate. This motivates their analysis which bounds the 2\to\infty distance, which is not quite entrywise, but better then l2. The bound also allows them to design a new stopping criteria which is more appropriate for applications which inspect the order of eigenvector entries. The authors present a nice set of applications and experiments exploring these applications.

Strengths: - Very nicely written paper. - Interesting results regarding the 2\to\infty errors in eigenvalue approximations. - A through discussion of the applications and nice experiments demonstrating the applicability of their results for these applications.

Weaknesses: - The relevance of the mixed 2->infty norm is unclear. Really, a maximum entrywise error is what you want, right? - A strange technical assumption (Assumption 2). Makes it feel there is an hole in the analysis. - The discussion of using approximation to realize the stopping criteria is somewhat hand waving. - Improvements in #iterations are nice but not really amazing.

Correctness: I did not read the proof carefully and verify them.

Clarity: Very well written paper.

Relation to Prior Work: There is very little discussion of previous work. Have entrywise bounds on approximate eigenvectors not been discussed before?

Reproducibility: Yes

Additional Feedback: - Line 34: Use of the \pm sign here can be confusing. You can use a \min. - Line 64, <X,Y> instead of <x,y> - Eq (3): are B and C of different dimensions? Isn't this supposed to be the seperation between two matrices (and thus of the same size?) - Instead of Assumption 1 you can define the coherence of a matrix (maximum row norm in an orthogonal basis) and state the results in terms of this coherence. I think this is better then imposing an "Assumption". - Lines 109-112: Why is the case that dist_2\to\infty << dist_2 the "typical case"? - Start of section 3: You use the terms "Ritz value" and "Ritz vector" without actually explaining what these are. - Eq (10): res_2\to\infty hasn't been actually defined. - Very hard to read Figure 4.


Review 2

Summary and Contributions: The paper uses the subspace iteration to compute the space spanned by top eigenvectors and establishes deterministic bounds for convergence when distances are measured in the l2→∞ norm. As a by-product, it proposes a practical stopping criterion and demonstrates its applicability via numerical experiments. The numerical results show that one can get comparable performance on downstream tasks while requiring fewer iterations by using the new stopping criterion.

Strengths: As the algorithm is well-known, the main contributions are the new upper bounds of the optimization error in the l2→∞ norm and the associated stopping criterion. The bounds are useful for understanding the performance in terms of the l2→∞ norm and give practical guidance on the stopping time. Given the wide applicability of the l2→∞ norm in various learning problems, the current study is timely and important.

Weaknesses: The technical derivations seem standard and not very difficult. The results are somehow expected. The improvements over the existing methods in terms of computation time seem limited.

Correctness: The technical proof seems correct, which is the main contribution. The comparisons of computation time are not fair in the numerical experiments. For example, for spectral clustering, t_comp is used to compute only the top eigenvector, whereas t_naive computes all eigenvectors. It will be more fair to compare computation for both algorithms only on the top eigenvector. But this would be the same algorithm (power iteration). Probably a fairer comparison is to fix j =1 in (13) and compares the time and accuracy.

Clarity: The technical results are clearly written and enjoyable to read. The norm ∥E∥_{2,2→∞} does not seem introduced. In the numerical experiments, more detailed descriptions are needed. For example, in Figure 1, is it log(metrics)? How many experiments? What are different ε representing, just running more iterations? In Figure 2, should be caption be "Distance (solid lines) and its approximations from equation (11) ...". You are not just talking about the rate here.

Relation to Prior Work: Yes, it is clearly discussed and the contributions are made clear.

Reproducibility: Yes

Additional Feedback:


Review 3

Summary and Contributions: This paper studies algorithms for computing eigenvalues/eigenvectors, and the uses of such routines to find important nodes in a graph. On the theoretical side it analyzes power iteration with a different convergence criteria based on the max difference at each eigenvalue. It then develops centrality and clustering algorithms using this computational primitive, and demonstrates moderate gains in both accuracy and performance.

Strengths: The power iteration method is extremely well studied, and the measure of convergence introduced here is novel compared to previous works. The experimental studies were extensive, and involves fairly large networks: it's a bit surprising to me that such gains can be extracted from an algorithm as classical as power iteration.

Weaknesses: It was difficult to find a new algorithmic idea compared to power iteration: the paper's contributions are mostly in the analysis and use of the eigenvectors computed. While the results obtained are good, it was a bit hard for me to see how to take these ideas further.

Correctness: Yes, both the bounds and experimental methodology are convincing. I do wish the experimental gains were compared more thoroughly against other, non-eigenvector based, methods such as ones based on matrix-factorizations/graph-embeddings, or graph neural networks.

Clarity: The paper is well written.

Relation to Prior Work: The conceptual differences in the bounds shown was clearly discussed. However, I'm less sure about the experimental comparisons with other methods for node identification / graph clustering.

Reproducibility: Yes

Additional Feedback: I learned quite a bit from this paper about additional properties of eigenvalue/vector computation routines, as well as new ways to apply such algorithms.


Review 4

Summary and Contributions: This paper analyzes the convergence of eigenvector decomposition of the subspace iteration when distances are measured in the l2→∞ norm and provide deterministic bounds.

Strengths: It provides an early stopping criterion based on theoretical analysis of convergence.

Weaknesses: First, I cannot check the proof details due to limited bandwidth and the fact that I am not an expert in this area. Second, as an Engineer, the impact of this work is pretty limited because spectral methods are not practical in industry. So improvement in this area is not attractive to me. The proof relies on a key assumption 2 at line 94. I don't see a reason why this assumption is true. This can be fatal for a theoretical paper. So I would suggest the author to develop practical algorithms showing faster convergence and making that is the focus of this paper.

Correctness: I'm not sure if the claim is correct because it's based on an assumption I'm not sure if it is correct. Empirical evaluation is not attractive to industrial practitioners.

Clarity: Yes.

Relation to Prior Work: Yes.

Reproducibility: No

Additional Feedback:

[Author Response · NeurIPS 2020]

We thank all of the reviewers for their time, careful reading, and valuable feedback.

Before discussing individual comments, we first want to address an issue raised by Reviewers 1 and 4 about **Assumption**
**2**. As we explain in the paper, Assumption 2 may not be necessary; pp. $7-9$ in the supplement contain a preliminary
convergence result in that direction. Also, it is *numerically verifiable*, as it need only hold for all $t$ elapsed by subspace
iteration. Indeed, we verify that the assumption holds for the datasets used in the experiments (see Section 1.3 of the
supplement).

**Reviewer 1.** We understand that lines 141–146 contain a somewhat hand-wavy discussion, as the formally correct
residual is given in eq. (9). However, this discussion is meant to provide practical alternatives for cases when we do not
know all information (e.g., incoherence or the exact value of sep). This is meant to mirror standard stopping criteria; for
example, criteria with respect to the spectral norm often assume that $\lambda_1 - \lambda_2 \approx \lambda_1$, and $\lambda_1 \approx \hat{\lambda}_1^t$.

The distinction between norms is another good point. The $\ell_{2\to\infty}$ norm is invariant to unitary transformations from the
right, while the $\ell_\infty$ norm is not. Thus, the $\ell_{2\to\infty}$ norm allows us to interpret a subspace geometrically, as in the case of
spectral clustering and this is a reason why the $\ell_{2\to\infty}$ norm is used in clustering analysis [3] (of course, when $r = 1$ the
two distances coincide). Also, for $E \in \mathbb{R}^{d\times r}$, we have that $\|E\|_\infty \le \|E\|_{2\to\infty} \le \sqrt{r}\,\|E\|_\infty$, which means $\ell_{2\to\infty}$ is a
good "proxy" for $\ell_\infty$. We can clarify these points when revising the paper.

Regarding coverage of prior work: as we mention in the manuscript, $\ell_{2\to\infty}$ norm convergence from a computational
perspective is absent from the literature (and this is a contribution of our paper) and prior research mostly focuses on
perturbation theory. We believe that we have done due diligence in citing such past work, but we are happy to include
any additional references that the reviewers think are relevant.

*A final clarification:* in lines 109–112, we write that $\mathrm{dist}_{2\to\infty} \ll \mathrm{dist}_2$ in the "typical" case, as this is only false when
the error is highly localized in just a few columns of the matrix; we will clarify this point in the paper.

**Reviewer 2.** One of the issues raised about the experiments may be due to a misunderstanding: $t_{\mathrm{naive}}$ checks $\|A\hat{V}_{:,j} -$
$\hat{\lambda}_j \hat{V}_{:,j}\| \le \varepsilon\hat{\lambda}_j$ for all $j$ **up to** $j = r$ (we will fix this in the updated manuscript). Therefore, $t_{\mathrm{naive}}$ does **not** compute
additional eigenvectors compared to $t_{\mathrm{comp}}$. We apologize for the confusion caused by that omission. We also appreciate
you bringing further issues to our attention, which will all be addressed in the updated manuscript, specifically:

• "computing $\|E\|_{2,2\to\infty}$" means computing $\|E\|_2$ and $\|E\|_{2\to\infty}$; it does not refer to a mixed norm.

• Figure 1 depicts how far apart the $\ell_2, \ell_{2\to\infty}$ distances and residuals are during an execution of subspace iteration,
with $\varepsilon$ representing different target accuracy levels, for a single experiment per configuration.

• In Figure 2, the quantities from Eq. (11) are hypothesized **rates** of convergence, rather than approximations to the
distance (e.g. in the sense of Eq. (10)), so we would prefer to leave this caption as is.

**Reviewer 3.** Limiting our experiments to eigenvector-based methods was intended, as the focus of the paper is on
theory and stopping criteria for eigensolvers and we want the experiments to highlight how our methods accelerate
computations of these existing methods. There are certainly alternatives to eigenvector-based methods, and we will
make sure to include this point when revising the paper. However, there are already several papers that discuss the
relative merits of these methods (e.g., of graph neural networks and spectral clustering) on downstream tasks. Thus, we
think it is more productive to cite this work, rather than duplicating the analysis. There are a few options for taking these
ideas further; for example, one could attempt a similar analysis for Krylov methods, such as the Lanczos algorithm. On
the other hand, more applications could benefit computationally from adopting appropriate stopping criteria (e.g., [4])
or analyzing algorithms from the $2 \to \infty$ perspective [1].

**Reviewer 4.** We understand that spectral methods are not used in some engineering settings. However, they are still the
basis of fundamental algorithms in machine learning and data science (e.g., PCA, factor analysis, robust statistics [2])
and an active area of applied research in technology companies such as Google[1] and StitchFix.[2]

## Footnotes

[1] https://ai.googleblog.com/2020/05/understanding-shape-of-large-scale-data.html

[2] https://www.wired.com/story/the-style-maven-astrophysicists-of-silicon-valley

# References

[1] M. Boedihardjo, S. Deng, and T. Strohmer. A Performance Guarantee for Spectral Clustering. *arXiv:2007.05627*, 2020.
[2] I. Diakonikolas and D. M. Kane. Recent advances in algorithmic high-dimensional robust statistics. *arXiv:1911.05911*, 2019.
[3] V. Lyzinski, D. Sussman, M. Tang, A. Athreya, and C. Priebe. Perfect Clustering for Stochastic Blockmodel Graphs via
Adjacency Spectral Embedding. *arXiv:1310.0532*, 2013.
[4] E. Nathan, G. Sanders, J. Fairbanks, V. E. Henson, and D. A. Bader. Graph ranking guarantees for numerical approximations to
katz centrality. *Procedia Computer Science*, 2017. International Conference on Computational Science.



[Meta-Review · NeurIPS 2020]

Overall the reviewers found that an improvement to such a fundamental, well-used algorithm, like the power method, was surprising and interesting, and the analysis for l2-linf norm was interesting. There were some weaknesses concerning the assumptions, but post-rebuttal the reviewers seemed to agree that Assumption 2 was not needed actually, and this caused a reviewer to think the proof was more complete / natural.